

# Limitations in Wavelet Analysis of Non-Stationary Atmospheric Gravity Wave Signatures in Temperature Profiles

Robert Reichert[1,2], Natalie Kaifler[2], and Bernd Kaifler[2]

[1]Meteorological Institute Munich, Ludwig-Maximilians-University, Munich, Germany
[2]Deutsches Zentrum für Luft- und Raumfahrt, Institut für Physik der Atmosphäre, Oberpfaffenhofen, Germany

**Correspondence:** Robert Reichert (robert.reichert@physik.uni-muenchen.de)

**Abstract.** Continuous Wavelet Transform (CWT) is a commonly used mathematical tool when it comes to the time-frequency (or distance-wavenumber) analysis of non-stationary signals and is used in a variety of research areas. In this work we use the CWT to investigate signatures of atmospheric internal gravity waves (GW) as observed in vertical temperature profiles obtained for instance by lidar. The focus is laid on the determination of vertical wavelengths of dominant GWs. According to

linear GW theory these wavelengths are a function of horizontal wind speed and hence, vertical wind shear causes shifts in the evolution of the vertical wavelength. The resulting signal fulfills the criteria of a chirp. Using complex Morlet wavelets, we apply CWT to test mountain wave signals modeling wind shears of up to $5\,\mathrm{ms}^{-1}\mathrm{km}^{-1}$ and investigate the capabilities and limitations. We find that the sensitivity of the CWT decreases for large chirp rates, i.e. strong wind shear. For a 4th order Morlet wavelet, edge effects become dominant at a vertical wind shear of $3.4\,\mathrm{ms}^{-1}\mathrm{km}^{-1}$. For higher-order wavelets,

edge effects dominate at even smaller values. In addition, we investigate the effect of GW amplitudes growing exponentially with altitude on the determination of vertical wavelengths. It becomes evident that, in case of conservative amplitude growth, spectral leakage leads to artificially enhanced spectral power at lower altitudes. Therefore, we recommend to normalize the GW signal before the wavelet analysis and the determination of vertical wavelengths. Finally, the cascading of receiver channels which is typical for middle atmosphere lidar measurements results in an exponential saw-tooth-like pattern of measurement

uncertainties as function of altitude. With the help of Monte Carlo simulations we compute a wavelet noise spectrum and determine significance levels, which enables the reliable determination of vertical wavelengths. Finally, the insights obtained from the analysis of artificial chirps are used to analyse and interpret real GW measurements from the Compact Rayleigh Autonomous Lidar in April 2018 at Río Grande, Argentina. The comparison of results of commonly used and our suggested wavelet analysis demonstrates improvements in the accuracy of determined wavelengths. For future analyses, we suggest the

usage of a 4th order Morlet wavelet, the normalization of GW amplitudes before wavelet analysis, and the significance level computation based on measurement uncertainties.

# 1   Introduction

The wavelet transform is a powerful mathematical tool to study non-stationary signals in time series and images. In contrast to the Fourier analysis, which decomposes a signal into a sum of sine and cosine functions, the wavelet analysis decomposes



the signal into a finite number of localized wavelets (Daubechies, 1990). It thus localizes signatures of interest in both time and frequency, making it a valuable tool for analyzing non-stationary signals. The wavelet transform has been used in a wide range of applications such as denoising (e.g. Pan et al., 1999; Alfaouri and Daqrouq, 2008; Tian et al., 2023), compression (e.g. Boix and Canto, 2010), feature extraction (e.g. Bruce et al., 2002; Seena and Yomas, 2014) and classification (e.g. Lambrou et al., 1998; Too et al., 2019) and is often applied to geophysical data (e.g. Torrence and Compo, 1998; Kaifler et al., 2017;

Bauer et al., 2020; Jin and Duan, 2021; Reichert et al., 2021). While the discrete wavelet transform is computationally cheap, it cannot capture the continuous time evolution of a signal. The continuous wavelet transform (CWT) provides a continuous representation of the signal in the time-frequency (or distance-wavenumber) domain, which makes it useful for analyzing signals that show time-dependent frequency variations.

  One example of such non-stationary signals are perturbations in air density and temperature caused by atmospheric internal

gravity waves (GW). These are localized and intermittent phenomena (Fritts and Alexander, 2003) that are generated due to e.g. flow over orography (e.g. Queney, 1948; Dörnbrack et al., 1999; Kaifler et al., 2015), propagating Rossby wave trains (e.g. Dörnbrack et al., 2022), and regions of strong wind shear (e.g. Fritts, 1982). Their spectral properties such as frequency and wavenumber are functions of atmospheric background conditions like stratification and wind shear which are rarely zero in the real atmosphere and hence, transient conditions most of the time result in non-stationary GW signals. The vertical wavelength

of stationary mountain waves (MW) excited by flow over orography is approximately given as $\lambda_z = 2\pi\frac{u}{N}$ where $u$ is the horizontal wind in the direction of wave propagation and $N$ is the thermal stability (Nappo, 2013). Since $u$ and $N$ are in most cases not constant in the real atmosphere, the vertical wavelength changes with altitude and time. In this work we investigate whether the CWT is a suitable tool to analyse the change in wavelength due to wind shears of different strengths.

  We focus on three major aspects of the CWT. First, one parameter that must be chosen is the non-dimensional frequency or

*order* $m_0$ of the wavelet. In case of the Morlet wavelet the order can be interpreted as the number of oscillations within the localization window. It determines the width of the wavelet in the time- and frequency-domain. A high order results in better frequency and worse time resolution, while the opposite is true for a low order. However, the order must not become arbitrary small as the admissibility condition must be fulfilled, i.e. the integral over the wavelet must be zero. In literature, there is no consensus on the optimal order of the Morlet wavelet. Many studies use an order of 6, as given in the widely cited work by

Torrence and Compo (1998). Other studies dealing with GW analysis use orders of 2, 4, 5 or 6.2 (see Table 1). However, the choice of the order plays an important role in the determination of vertical wavelengths as these can change rapidly depending on vertical wind shear.

  Second, according to linear GW theory, not only vertical wavelengths can change quickly but also amplitudes of GWs increase exponentially with altitude enforced by conservation of energy in the absence of dissipation and decreasing air density. This

rapid increase in GW amplitude might induce an unwanted shift in the localization of the wavelet during the computation of the CWT. To our knowledge, only few studies have normalized their GW signals before the wavelet analysis in order to prevent amplitude growth-induced errors (Wright et al., 2017; Vadas et al., 2018; Gisinger et al., 2022).



| Publication | Latitude | Height range | $T'$ separation method | Significance | COI | $m_0$ | Uncertainties | Range of $\lambda_z$ |
|---|---|---|---|---|---|---|---|---|
| (Chane-Ming et al., 2000) | 21°S | 30-60 km | Temporal mean subtraction and vertical Butterworth filtering | $T' > 1$ K | - | 5.336 | up to 0.86 % | 1-10 km |
| (Werner et al., 2007) | 69°N | 30-60 km | Polynomial fit | 50 % of $T'$ maximum | - | 6.2 | - | 3-8.5 km |
| (Rauthe et al., 2008) | 54°N | 1-105 km | Nightly mean subtraction | - | yes | 5 | 1.5-2.5 K, never exceed 10 K | 6-48 km |
| (Ehard et al., 2016) | 68°N | 30-65 km | Sliding cubic spline | - | yes | 6 | - | 7-12 km |
| (Baumgarten et al., 2017) | 54°N | 30-70 km | Daily mean subtraction and Butterworth filtering | - | yes | - | - | 3-20 km |
| (Reichert et al., 2021) | 54°S | 15-95 km | subtraction of long-period subseasonal oscillations | Spectral power >50 % of $T$ uncertainty | yes | 4 | 0.3-10 K | 4-30 km |
| (Wing et al., 2021) | tropical | 15-95 km | Nightly mean subtraction and Butterworth filtering | - | yes | - | 0.1 K, never exceed 10 % | 4-7 km |
| (Gisinger et al., 2022) | 54°S | 15-80 km | Butterworth filtering | - | yes | 2 | | 2-15 km |

**Table 1.** Overview of publications dealing with vertical wavelength determination in lidar temperature soundings using the CWT. Every study has used the complex Morlet wavelet in their analysis.

Table 1 lists publications that determine vertical wavelengths of GWs based on wavelet analysis. While the discussion on background and perturbation separation is a common one and has been summarized, for instance, by Ehard et al. (2015), we seek to establish guidelines on best practices for the determination of vertical wavelengths. We note that all listed works address measurement uncertainties and significance levels in the wavelet power spectrum (WPS) differently, if at all. Also, the cone of influence (COI) that indicates the region where edge effects may influence the WPS is not dealt with or not even mentioned in some papers. Therefore, as the third and last aspect of our work, we address the propagation of measurement uncertainties into the WPS, the computation of significance levels and hence the reliability of determined vertical wavelengths.

Ultimately, it is important to determine the vertical wavelength of GWs correctly since this parameter provides valuable information on the dynamics of the mean-flow. A shrinking vertical wavelength, for instance, may be indicative of a reduction in horizontal wind speed and, in extreme cases when the wavelength approaches zero, may point to a critical level, i.e. a level where the intrinsic phase speed of a GW becomes equal to the horizontal wind speed and GW dissipation occurs. In addition, the vertical wavelength is a crucial quantity in raytracing (Marks and Eckermann, 1995; Geldenhuys et al., 2021) and is used in the computation of GW momentum fluxes (e.g. Ern et al., 2022). Moreover, knowledge about the vertical wavelength is necessary to derive temperature amplitudes and momentum fluxes in the mesosphere lower thermosphere from OH-airglow observations (Fritts et al., 2014).





Our considerations culminate in the following three questions:

1. What is the optimal choice for the order of the Morlet wavelet given a considerable vertical wind shear that gives rise to shifts in vertical wavelength of GWs?

2. Assuming a conservative growth rate of GW amplitudes with altitude, what is the benefit of normalizing GW amplitudes before applying the wavelet analysis?

3. How do measurement uncertainties affect the results of wavelet analysis and, in particular, which parts of the WPS can 80 be trusted to representing reliable power estimates?

This publication is structured as follows. In Section 2 we first give a brief repetition on the mathematical foundation of the CWT and define four linear chirps as test signals. After that we investigate the research questions in Section 3 based on the defined test signals and, subsequently, we present a case study demonstrating the application of wavelet analysis to GW signatures observed by the CORAL lidar in Argentina. Section 4 discusses the results and Section 5 gives a summary, conclusions, and 85 recommendations on how to determine vertical wavelengths of non-stationary GW signatures in the middle atmosphere in the form of short step-by-step instructions.

## 2 Methods

### 2.1 Continuous Wavelet Transform

In the following we will recall the building blocks of the CWT and introduce the commonly used terms. The interested reader 90 is referred to Torrence and Compo (1998); Maraun and Kurths (2004); Ge (2008) for detailed information.

The core of the CWT is the mother wavelet that is in this work chosen to be the complex Morlet wavelet and given as

$$\psi_0(\eta) = \pi^{-1/4} e^{im_0\eta} e^{-\eta^2/2} \tag{1}$$

where $\eta$ is the non-dimensional length and $m_0$ is the non-dimensional wavenumber. $m_0$ is also called order of the Morlet wavelet. Morlet wavelets are a class of wavelets that are commonly used in geophysics (e.g. Grinsted et al., 2004; Wong et al., 95 2012; Kaifler et al., 2017; Llamedo et al., 2019; Wu et al., 2021; Reichert et al., 2021; Geldenhuys et al., 2022). Daughter wavelets are scaled ($s$) and translated ($\xi$) versions of the mother wavelet such that

$$\psi(\xi, s) = c(s)\psi_0\left(\frac{z - \xi}{s}\right), \tag{2}$$

where $z$ is altitude and $c(s)$ is a normalization factor. Normalization can be performed in two different ways: Either one requires a flat white noise spectrum or sines of the same amplitude having the same integrated power in the wavenumber 100 domain. Torrence and Compo (1998) have defined $c(s) = \sqrt{\frac{\delta z}{s}}$ which results in a flat white noise spectrum but sines of same amplitude have less spectral power at larger scales. In order to allow for a fair comparison of peaks in the WPS we follow





Maraun and Kurths (2004) and references therein and define $c(s) = \sqrt{\delta z}$, where $\delta z$ is the vertical sampling interval.

The CWT of a temperature signal $T(z)$ is given by the convolution with the set of daughter wavelets:

$$W(z,s) = c(s) \int_{z_i}^{z_f} T(z)\psi^*(z - \xi, s) d\xi \tag{3}$$

$$= c(s) \int_{-m_{max}}^{m_{max}} \hat{T}(m)\hat{\psi}^*(m,s)e^{imz}dm, \tag{4}$$

where $z_i$ and $z_f$ define the altitude range of the measurement, $(^*)$ indicates the complex conjugate, $(\hat{\ })$ indicates the Fourier transform, and $m_{max} = \frac{1}{2\delta z}$. Equation 4 makes use of the convolution theorem. As the Morlet wavelet is complex, also the wavelet transform is complex and the WPS is defined as $|W(z,s)|^2$.

The scales of the daughter wavelets are computed as

$$s_j = s_0 2^{j\delta j}, \quad j = 0, 1, ..., J \tag{5}$$

$$J = \frac{1}{\delta j} log_2\left(\frac{N\delta z}{s_0}\right), \tag{6}$$

where $s_0$ is the smallest resolvable scale and set to $s_0 = 2\delta z$, $\delta j = \frac{1}{16}$, and $J$ determines the largest scale which depends on the altitude range of the measurement.

In case of non-periodic signals, the computed spectral power at the edges of the WPS, i.e. where $z - z_i < \sqrt{2}s$ or $z_f - z < \sqrt{2}s$, where $z_i$ and $z_f$ are starting and ending altitudes respectively, is not reliable and might be overestimated. The factor $\sqrt{2}s$ is called $e$-folding time (in our context $e$-folding length) and ensures that spectral power from edge discontinuities drops by a factor of $e^{-2}$. This $e$-folding length region where the spectral power at the edges of the WPS is affected by discontinuities is commonly referred to as the cone of influence (COI). It is suggested to pad the profile with zeros up to the next power of two before applying the CWT. This results in an underestimation of spectral power at the edges and a speed up of the FFT algorithm.

The chosen scales are not necessarily identical to wavelengths. The conversion is given by

$$\lambda = \frac{2\pi s}{m_0 + \sqrt{2 + m_0^2}}. \tag{7}$$

A scale of $s = 1\,\mathrm{km}$ is equivalent to a wavelength in the range from $1.5\,\mathrm{km}$ to $0.6\,\mathrm{km}$ for the order ranging from four to ten. Werner et al. (2007) have chosen an order of 6.2 which ensures that scales and wavelengths are identical. In the next sections we define four linear chirps $LC_i(z)$ that serve as test signals with well-known wavelengths $\lambda_{\text{input}}(z)$, amplitudes $a(z)$, and noise levels $r(z)$. Subsequently, the CWT of $LC_i(z)$ are computed and the locations of the maxima in the WPS are used to determine the $\lambda_{\text{output}}(z)$ as function of altitude. The ratio of output to input wavelength is used as a metric to quantify how well the wavelet analysis has captured the chirp.





## 2.2 Definition of test signals

To assess the performance of the wavelet analysis we define four linear chirps according to

$$LC_i(z) = a_i(z) \sin\left(2\pi \int_0^z \frac{1}{\lambda_i(\tilde{z})} d\tilde{z}\right) + r_i(z) \tag{8}$$

where $a_i(z)$ are amplitudes, $r_i(z)$ are random numbers from a Gaussian distribution with mean $\mu = 0$ and standard deviation $\sigma_i$, and the vertical wavelengths $\lambda_i$ are given as

$$\lambda_i(z) = \frac{2\pi \partial_z u_i}{N} z + \lambda_0, \tag{9}$$

where $N = 0.02\,\mathrm{s}^{-1}$, $\lambda_0 = 0.2\,\mathrm{km}$, and $z$ ranges from $0\,\mathrm{km}$ to $100\,\mathrm{km}$. The first two chirps, $LC_1$ and $LC_2$, have constant amplitudes of $a_{1,2} = 1.0\,\mathrm{K}$, no additive white Gaussian noise $\sigma_{1,2} = 0.0\,\mathrm{K}$, and differ only by the chirp rates which are computed from wind shears of $\partial_z u_1 = 5\,\mathrm{ms}^{-1}\mathrm{km}^{-1}$ and $\partial_z u_2 = 2.5\,\mathrm{ms}^{-1}\mathrm{km}^{-1}$, which are typical values for a real atmosphere (Fig.1ab). The latter two chirps, $LC_3$ and $LC_4$, have a chirp rate according to a wind shear of $\partial_z u_{3,4} = 2.5\,\mathrm{ms}^{-1}\mathrm{km}^{-1}$, show a linear amplitude growth according to $a_{3,4}(z) = 1.0\,\mathrm{K} + z\,\mathrm{Kkm}^{-1}$ but differ in their additive Gaussian white noise levels. While $LC_3$

has no additive noise, $LC_4$ shows a constant noise level a standard deviation of $\sigma_4 = 5\,\mathrm{K}$, and uncorrelated values every $100\,\mathrm{m}$ (Fig.1cd). An overview of the chirp parameters is given in Table 2.

|  | $a\,/\,\mathrm{K}$ | $\partial_z u\,/\,\mathrm{ms}^{-1}\mathrm{km}^{-1}$ | $\sigma\,/\,\mathrm{K}$ |
|---|---|---|---|
| $LC_1$ | 1.0 | 5.0 | 0.0 |
| $LC_2$ | 1.0 | 2.5 | 0.0 |
| $LC_3$ | $z\,\mathrm{km}^{-1} + 1.0$ | 2.5 | 0.0 |
| $LC_4$ | $z\,\mathrm{km}^{-1} + 1.0$ | 2.5 | 5.0 |

**Table 2.** Parameter of the four defined linear chirps.





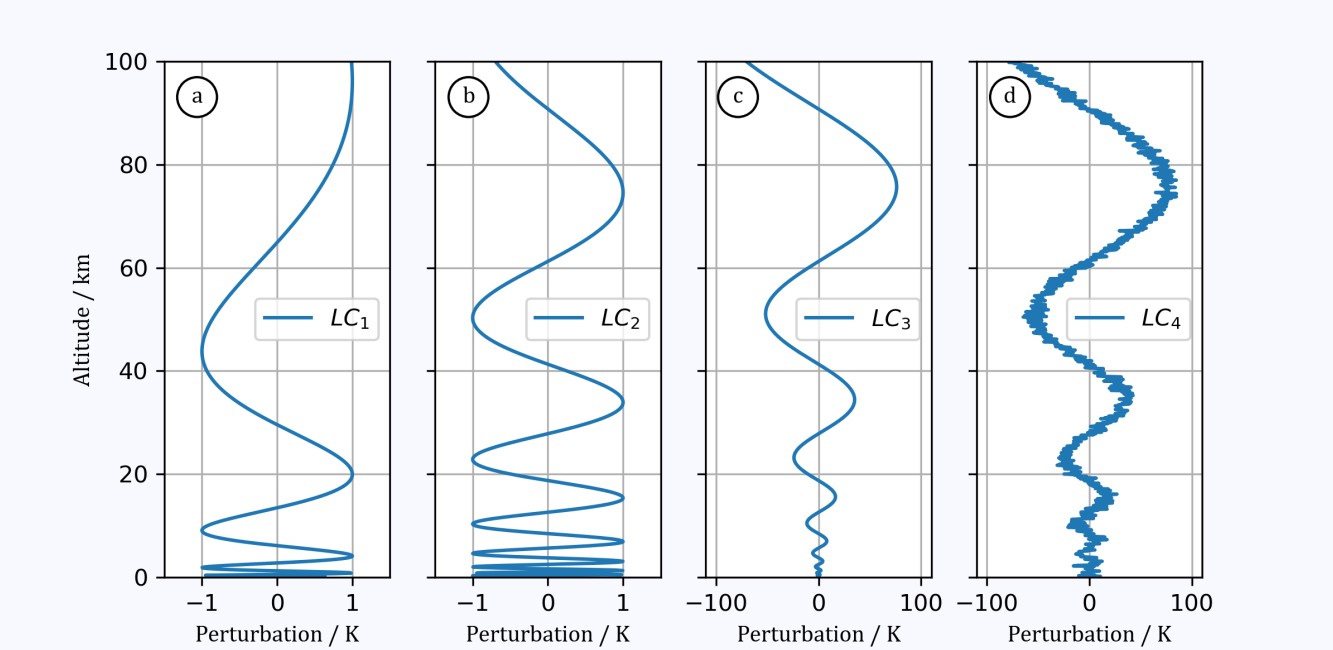

**Figure 1.** Four defined linear chirps with either constant (ab) or linearly growing (cd) amplitudes. Wavelengths derived from wind shears of $5\,\mathrm{ms^{-1}km^{-1}}$ (a) and $2.5\,\mathrm{ms^{-1}km^{-1}}$ (bcd) change linearly. The added Gaussian white noise is $\sigma = 0\,\mathrm{K}$ (abc) and $\sigma = 5\,\mathrm{K}$ (d).

# 3 Results

## 3.1 What is the optimal choice for the order of the Morlet wavelet?

Figure 2 illustrates the WPS of $LC_1$ and $LC_2$ using wavelet orders of 4 and 6. We focus on the lowermost $50\,\mathrm{km}$ since vertical
wavelengths increase further above and the WPS maximum lies completely in the COI there. When using a 6th order wavelet, we find that the WPS maximum of $LC_2$ is entirely within the COI (Fig. 2a). Comparing the evolution of input and output wavelengths, we find mostly good agreement but also one discontinuity at $35\,\mathrm{km}$. This discontinuity disappears when using a 4th order wavelet (Fig. 2b), resulting in the WPS maximum being now outside the COI. We notice that output wavelengths are at all altitudes shorter than input wavelengths. When increasing the wind shear to $5\,\mathrm{ms^{-1}km^{-1}}$, the WPS maximum of $LC_1$ is
located within the COI and again a discontinuity occurs (Fig. 2c).





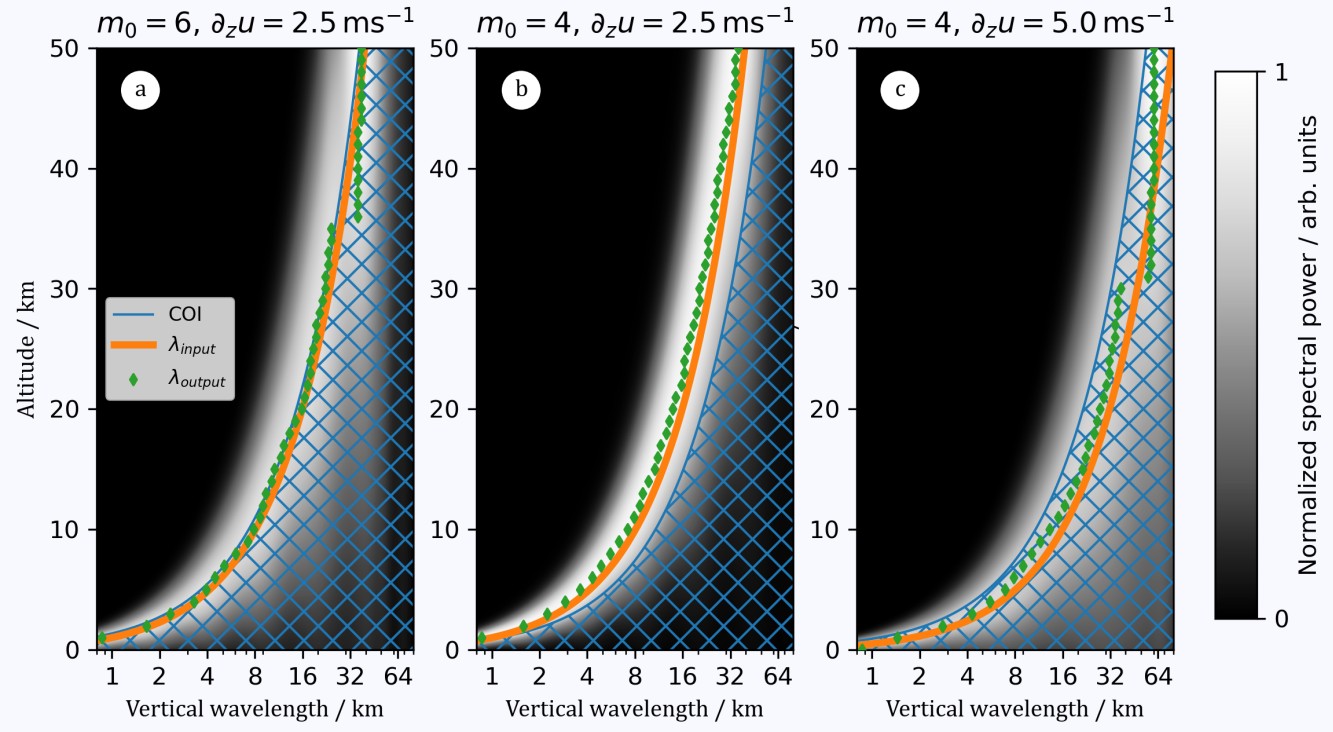

**Figure 2.** WPS of $LC_2$ using $m_0 = 6$ (a), $m_0 = 4$ (b), and WPS of $LC_1$ using $m_0 = 4$ (c). Orange lines mark the input wavelength as function of altitude and green diamonds mark the determined output wavelength, i.e. the maximum in the WPS at each $z$. The hatched blue regions mark the COI.

To quantify the level of agreement between input and output wavelength we compute their ratio as function of wind shear and wavelet order. For that we generate more linear chirps similar to $LC_1$ and $LC_2$ with constant amplitudes, no additive noise, and wavelengths computed from wind shears in the range $0.5 - 4.5\,\mathrm{ms}^{-1}\mathrm{km}^{-1}$. We compute the WPS using wavelet orders of 4, 6, and 8 and determine the output wavelengths as the WPS maxima. Figure 3 illustrates the distributions of wavelength ratios

derived from the lowermost $50\,\mathrm{km}$ of the simulated altitude range and Table 3 lists the corresponding median deviations as well as interquartile ranges (IQR). We find an average negative deviation from the input wavelength of $\approx -10\,\%$ for $m_0 = 4$, while the IQR stays below $10\,\%$ for all considered wind shears. For wavelet orders of 6 and 8 we notice smaller median deviations from unity, while the IQR exceeds $10\,\%$ starting at a wind shear of $3.0\,\mathrm{ms}^{-1}\mathrm{km}^{-1}$ and $2.0\,\mathrm{ms}^{-1}\mathrm{km}^{-1}$, respectively. The increase in IQR is most likely due to increasing edge effects such as mentioned discontinuities shown in Figure 2ac. The

$e$-folding line marking the COI can also be understood as a chirp rate that corresponds to a maximum wind shear up until edge effects are negligible. Assuming a constant $N = 0.02\,\mathrm{s}^{-1}$ for the case $m_0 = 4$, the maximum wind shear that is reached until edge effects can be considered minor is $3.4\,\mathrm{ms}^{-1}\mathrm{km}^{-1}$, for $m_0 = 6$ it is $2.3\,\mathrm{ms}^{-1}\mathrm{km}^{-1}$, and for $m_0 = 8$ it is $1.8\,\mathrm{ms}^{-1}\mathrm{km}^{-1}$. These values are in line with the broadening of the wavelength ratio distributions in Figure 3 and the notable increase in IQRs in Table 3.

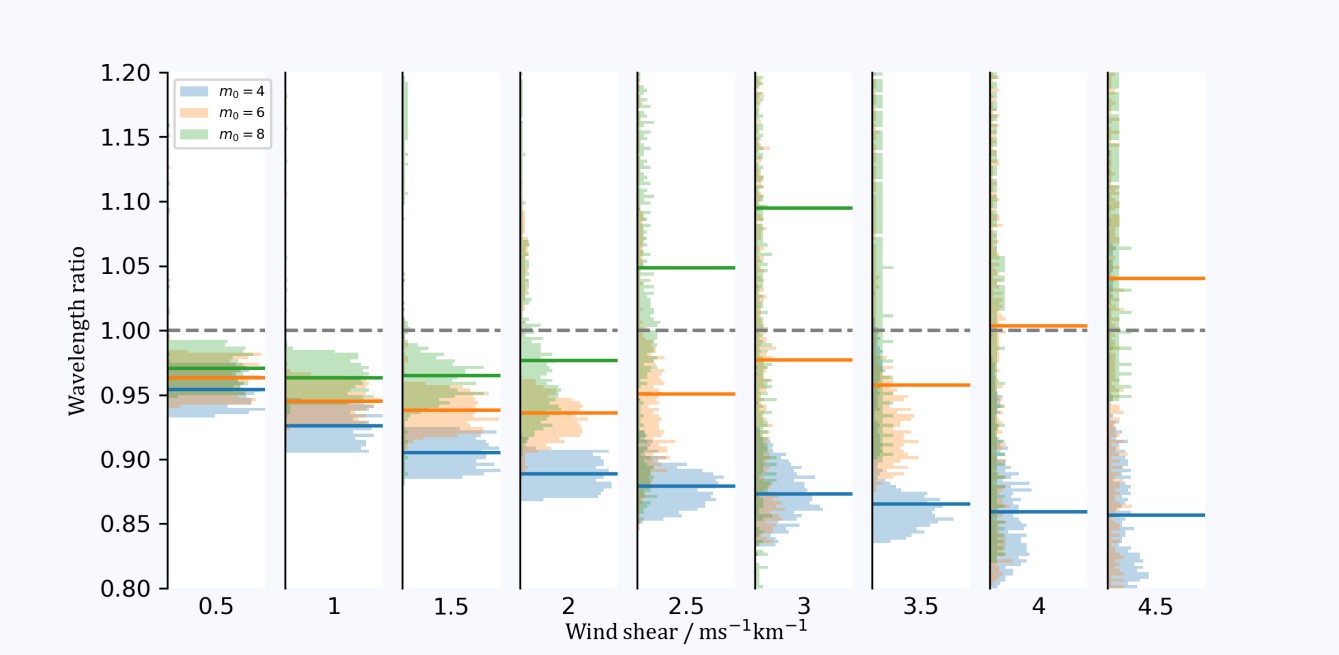

**Figure 3.** Normalized histograms of the wavelength ratio as function of wind shear and wavelet order (blue: $m_0 = 4$, orange: $m_0 = 6$, green: $m_0 = 8$). The determination of wavelengths is based on linear chirps with constant amplitude and no additive noise. Horizontal lines represent the median of the distribution. Note that for $m_0 = 8$ and wind shear $> 3.5\,\mathrm{ms}^{-1}\mathrm{km}^{-1}$ the median lies outside the plot range. The grey dashed line marks a wavelength ratio of one. See Table 3 for details.

| $\partial_z u/\mathrm{ms}^{-1}\mathrm{km}^{-1}$ | 0.5 | 1.0 | 1.5 | 2.0 | 2.5 | 3.0 | 3.5 | 4.0 | 4.5 |
|---|---|---|---|---|---|---|---|---|---|
| $m_0 = 4$ | $-4.6^{+1.1}_{-1.1}$ | $-7.4^{+1.0}_{-1.1}$ | $-9.5^{+1.0}_{-1.0}$ | $-11.1^{+1.0}_{-1.0}$ | $-12.1^{+1.0}_{-1.1}$ | $-12.7^{+1.7}_{-1.4}$ | $-13.5^{+2.7}_{-1.2}$ | $-14.1^{+3.0}_{-2.7}$ | $-14.3^{+5.3}_{-4.3}$ |
| $m_0 = 6$ | $-3.7^{+1.1}_{-1.1}$ | $-5.5^{+1.1}_{-1.0}$ | $-6.2^{+1.1}_{-1.0}$ | $-6.4^{+1.6}_{-1.3}$ | $-4.9^{+4.3}_{-3.8}$ | $-2.2^{+12.9}_{-8.9}$ | $-4.2^{+26.0}_{-3.9}$ | $0.4^{+15.1}_{-10.8}$ | $4.0^{+23.1}_{-14.8}$ |
| $m_0 = 8$ | $-3.0^{+1.1}_{-1.1}$ | $-3.7^{+1.0}_{-1.0}$ | $-3.5^{+1.4}_{-1.4}$ | $-2.3^{+8.3}_{-2.8}$ | $4.8^{+12.7}_{-7.0}$ | $9.6^{+24.4}_{-15.1}$ | $80.0^{+>100}_{-59.8}$ | $57.2^{+>100}_{-51.7}$ | $89.0^{+>100}_{-62.9}$ |

**Table 3.** Median deviation from a wavelength ratio of unity and IQR in percent as function of wind shear and wavelet order. Green cells mark distributions with an IQR smaller than $10\,\%$.

## 3.2 What is the benefit of normalizing GW amplitudes before applying the wavelet analysis?

It is expected that GW amplitudes grow exponentially with altitude in the absence of dissipation due to conservation of energy since air density decreases with altitude. Do amplitudes growing with altitude affect the determination of vertical wavelengths in the wavelet analysis? To investigate potential effects, we consider the general solution to the Taylor-Goldstein equation for $u = 0\,\mathrm{ms}^{-1}$ and without loss of generality at $x = t = 0$. Furthermore, for simplicity, we limit our analysis to the solution for an





upward propagating wave. See also equations 2.54-2.56 in Nappo (2013). The GW's temperature signature is given as

$$T(z) = T_0 e^{z/2H} e^{im_0 z/s},$$                                                                           (10)

where $T_0$ is an arbitrary initial temperature and $H$ is the density scale height that is in the middle atmosphere in the order of 6-8 km. The product of $T(z)$ with the complex conjugate daughter Morlet wavelet at $\xi = 0$ evaluates to

$$T(z)\psi^*(z,s) = T_0 e^{z/2H} e^{im_0 z/s} \cdot \pi^{-1/4} e^{-im_0 z/s} e^{-z^2/2s^2}$$                         (11)

175                  $$= T_0 \pi^{-1/4} e^{z/2H} e^{-z^2/2s^2}.$$                                                  (12)

The position of the peak of this product is not in agreement with the peak of the Morlet wavelet's Gaussian window anymore but is located at $z = \frac{s^2}{2H}$. It becomes clear that during the computation of the convolution (Equ. 4), due to the exponential growth of GW amplitudes, spectral power leaks from $z = z_0 + \frac{s^2}{2H}$ down to $z = z_0$. In other words, spectral amplitudes computed at $(z, s)$ are dominated by wave amplitudes at $\left(z + \frac{s^2}{2H}, s\right)$. For example, assuming $H = 7\,\text{km}$ and a vertical wavelength of

$\lambda_z = 2\,\text{km}$, spectral leakage occurs over an altitude range of $0.5\,\text{km}$, while for a vertical wavelength of $\lambda_z = 10\,\text{km}$ spectral power leaks over an altitude range of 12 km. Figure 4 illustrates the consequences in case the wavelength is determined without normalizing the amplitudes before the wavelet analysis. Output wavelengths identified as the maxima in the WPS of $LC_3$ deviate significantly from input wavelengths when amplitudes grow linearly (Fig. 4b). As mentioned above, due to spectral leakage, the WPS at low altitudes is strongly affected by large-scale large-amplitude signals at higher altitudes. Please note

that increasing amplitudes affect the determination of wavelengths only in case the wavelength also changes with altitude.
        We investigate whether a normalization of the signal before the wavelet analysis can mitigate the problem of spectral leakage. We suggest the following: First, we compute the running root-mean-square (RMS) of the temperature perturbations over a boxcar window of length $L$. Ideally, $L$ covers one vertical wavelength. A multiplication with a factor or $\sqrt{2}$ converts the running RMS of temperature perturbations into what can be considered GW amplitudes. Second, we fit a 4th degree polynomial to the

derived GW amplitudes. Finally, we normalize the temperature perturbations by dividing them by the result of the polynomial fit. Following this procedure, the output wavelengths retrieved from the WPS are again in reasonable agreement with the input wavelengths as demonstrated in Figure 4c.





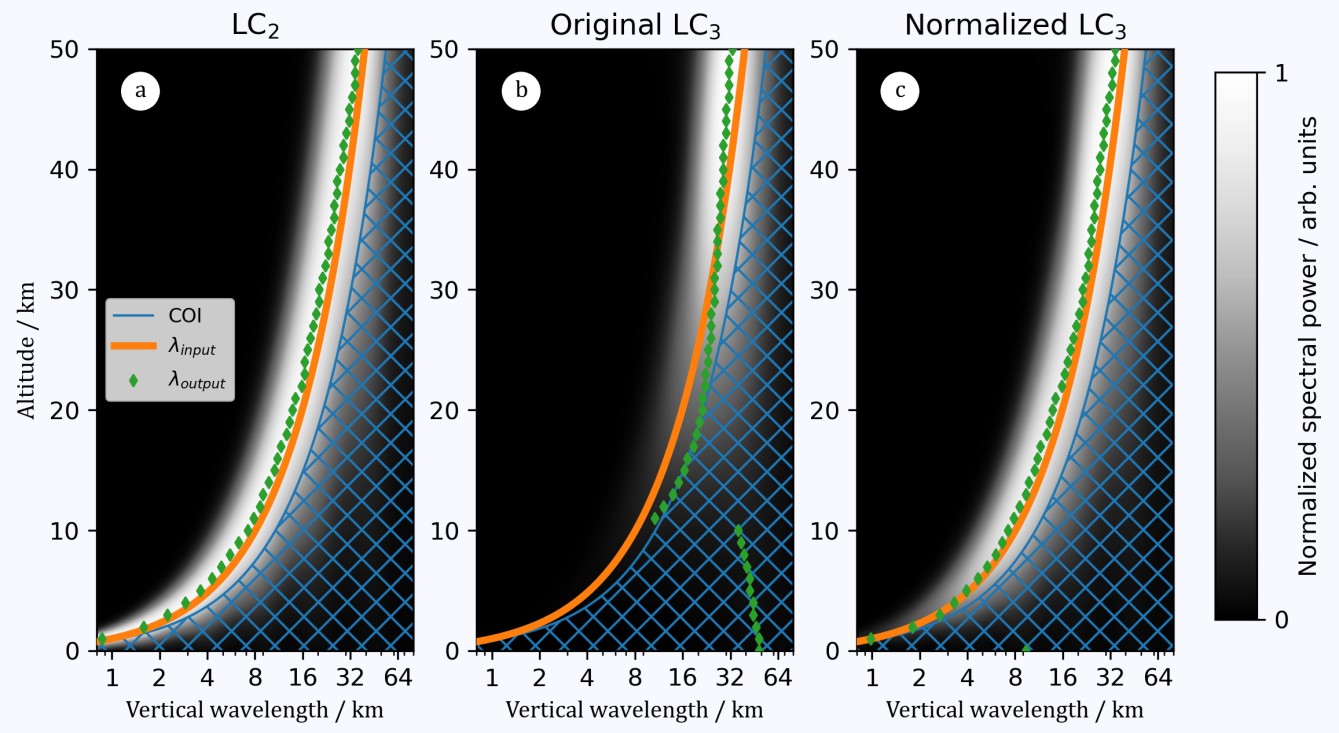

**Figure 4.** WPS of $LC_2$ (a), WPS of $LC_3$ (b), and WPS of $LC_3$ after amplitude normalization for $m_0 = 4$ (c). Orange lines mark the input wavelength and green diamonds mark determined output wavelengths, i.e. the maximum in the WPS at each $z$. The hatched blue regions mark the COI.

In order to further quantify the effect of growing amplitudes on the determination of vertical wavelengths, we multiply the linear chirps from Section 3.1 with linearly increasing amplitudes according to growth rates of $0.1\,\mathrm{K\,km^{-1}}$, $1\,\mathrm{K\,km^{-1}}$, and $10\,\mathrm{K\,km^{-1}}$. The modified chirps are analysed with and without normalization using a 4th order wavelet. Distributions of wavelength ratios are illustrated in Figure 5 and median deviations from a wavelength ratio of unity as well as IQRs are given in Table 4.



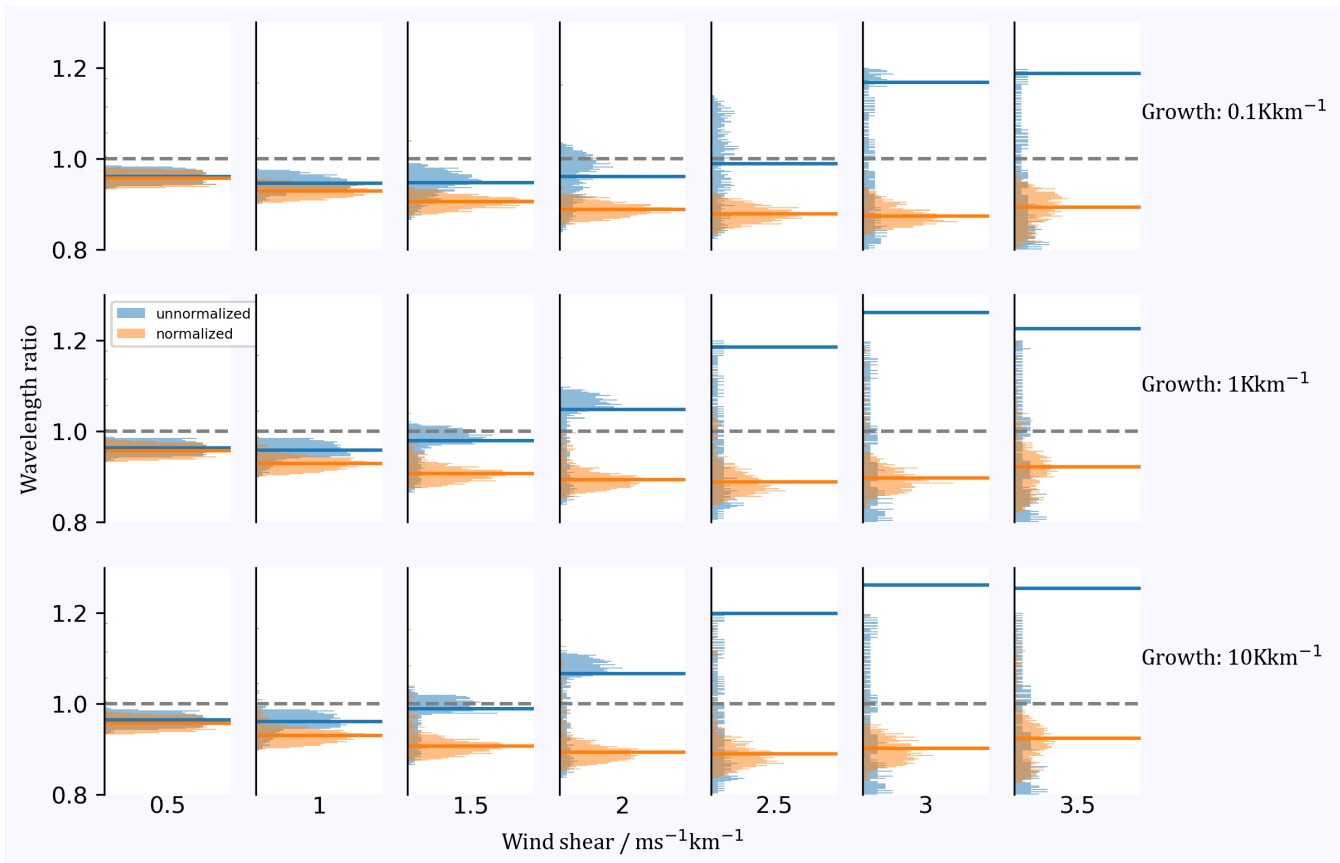

**Figure 5.** Normalized histograms of the wavelength ratio as function of wind shear and amplitude growth rate (blue: without normalization, orange: with normalization). The wavelet order is $m_0 = 4$. Horizontal lines represent the median of the distribution. The grey dashed lines mark a wavelength ratio of one. See Table 4 for details.

| Growth rate | $\partial_z u/\mathrm{ms}^{-1}\mathrm{km}^{-1}$ | 0.5 | 1.0 | 1.5 | 2.0 | 2.5 | 3.0 | 3.5 |
|---|---|---|---|---|---|---|---|---|
| 0.1 Kkm$^{-1}$ | not normalized | $-3.9^{+1.1}_{-1.0}$ | $-5.4^{+1.2}_{-1.1}$ | $-5.3^{+1.5}_{-2.2}$ | $-3.9^{+3.1}_{-4.5}$ | $-1.1^{+6.5}_{-7.7}$ | $16.9^{+22.4}_{-24.7}$ | $18.7^{+>100}_{-28.1}$ |
| | normalized | $-4.4^{+1.1}_{-1.0}$ | $-7.2^{+1.0}_{-1.0}$ | $-9.5^{+0.9}_{-1.0}$ | $-11.2^{+1.1}_{-1.1}$ | $-12.2^{+1.3}_{-1.3}$ | $-12.6^{+1.7}_{-1.3}$ | $-10.7^{+2.4}_{-2.7}$ |
| 1 Kkm$^{-1}$ | not normalized | $-3.6^{+1.0}_{-1.1}$ | $-4.2^{+1.2}_{-1.2}$ | $-2.1^{+1.5}_{-2.7}$ | $4.8^{+2.3}_{-10.2}$ | $18.5^{+19.3}_{-24.8}$ | $26.4^{+>100}_{-33.8}$ | $22.5^{+>100}_{-31.2}$ |
| | normalized | $-4.3^{+1.0}_{-1.0}$ | $-7.1^{+1.0}_{-1.1}$ | $-9.3^{+1.3}_{-1.4}$ | $-10.7^{+1.7}_{-1.4}$ | $-11.2^{+2.6}_{-1.8}$ | $-10.3^{+2.7}_{-1.9}$ | $-7.9^{+3.8}_{-3.6}$ |
| 10 Kkm$^{-1}$ | not normalized | $-3.5^{+1.0}_{-1.1}$ | $-3.9^{+1.3}_{-1.2}$ | $-1.1^{+1.6}_{-3.4}$ | $6.6^{+2.5}_{-11.8}$ | $19.9^{+67.5}_{-26.2}$ | $26.4^{+>100}_{-33.8}$ | $25.5^{+>100}_{-34.1}$ |
| | normalized | $-4.3^{+1.0}_{-1.0}$ | $-7.0^{+1.1}_{-1.2}$ | $-9.3^{+1.4}_{-1.2}$ | $-10.6^{+1.8}_{-1.4}$ | $-11.0^{+2.7}_{-1.9}$ | $-9.8^{+2.9}_{-2.1}$ | $-7.5^{+4.3}_{-3.5}$ |

**Table 4.** Median deviation from a wavelength ratio of one and IQR in percent as function of wind shear and growth rate. Green cells mark distributions with IQRs smaller than $10\%$.





We notice that for wind shears less than $1.5\,\mathrm{ms}^{-1}\mathrm{km}^{-1}$ the wavelength ratio distributions deviate less from unity when no normalization is applied regardless of the amplitude growth rate. However, when wind shears exceed $2\,\mathrm{ms}^{-1}\mathrm{km}^{-1}$, we find

that the IQRs for not normalized chirps increase drastically while the normalization keeps the IQRs at a low level.

### 3.3 How do measurement uncertainties affect the results of wavelet analysis and, in particular, which parts of the WPS can be trusted to representing reliable power estimates?

Every measurement is subject to measurement uncertainties. To model a simple case, we assume a white noise spectrum. To distinguish a physically meaningful signal from noise, we need to know how the noise is reflected in the WPS. We propose the

following approach: First, we generate 5,000 Gaussian white-noise profiles with a vertical resolution of $100\,\mathrm{m}$, with $\mu = 0\,\mathrm{K}$, and $\sigma = 5\,\mathrm{K}$ and compute the WPS of each noise profile. From the set of 5,000 WPS we determine the 99th percentile of spectral power as function of $z$ and $\lambda_z$. Any spectral power in the signal's WPS above this 99th percentile is considered to be significant on the $99\,\%$ level.

We now inspect the WPS of $LC_4$ (the linear chirp with growing amplitudes and added noise, Fig. 6a) and find a similar

distribution of wavelengths as for $LC_3$ (the linear chirp with growing amplitudes without added noise, Fig. 4b). To mitigate the problem of spectral leakage due to growing wave amplitudes, we normalize $LC_4$ as described in Section 3.2 and obtain the results shown in Figure 6b. As evident from Figure 6b, the normalization results in an increase of spectral power of the linear chirp but also of noise notable below $10\,\mathrm{km}$. In other words, where the signal-to-noise ratio is low (in this case below 2) the determination of vertical wavelengths is not reliable. It is crucial to compute significance levels to determine physically

meaningful wavelengths.





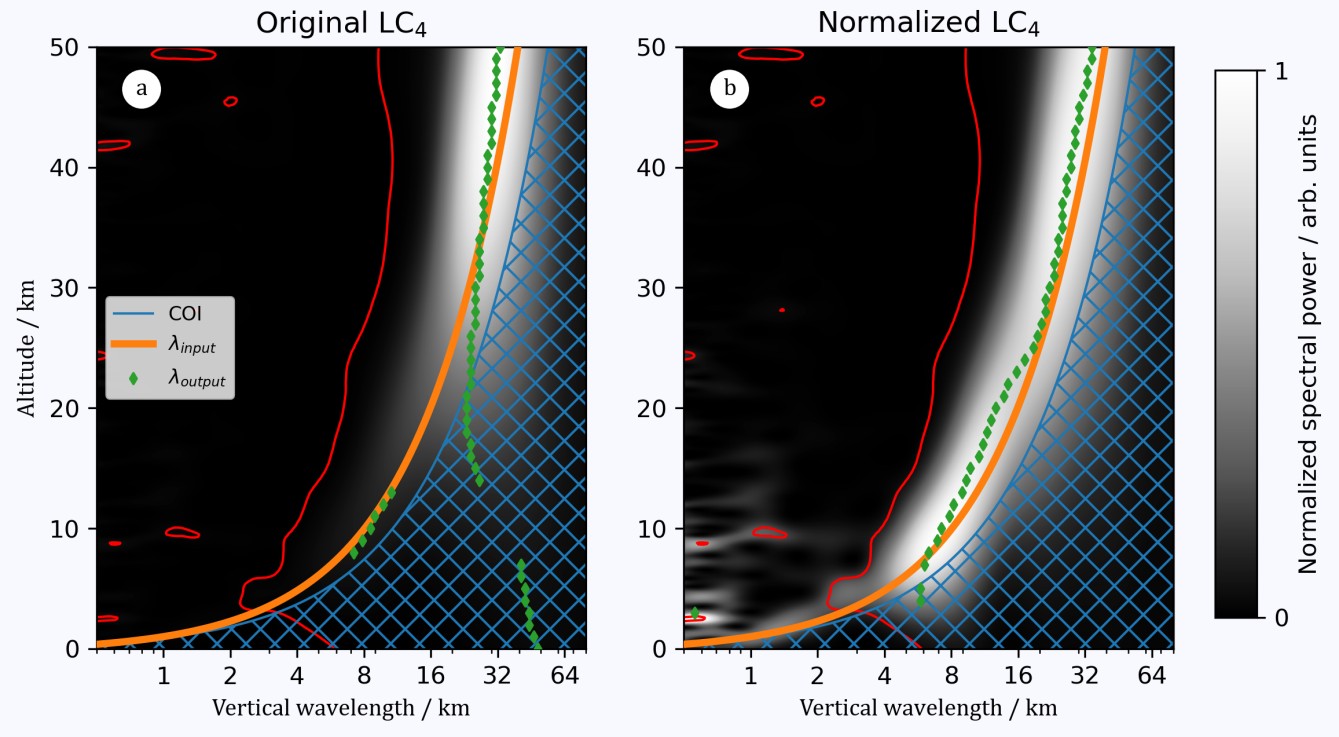

**Figure 6.** WPS of the original $LC_4$ (a) and of the normalized $LC_4$ (b). $m_0 = 4$ was used in the CWT. Red contours mark the $99\%$ significance level. Orange lines mark the input wavelength and green diamonds determined output wavelengths, i.e. the maximum in the WPS at each $z$. The hatched blue regions mark the COI.

### 3.4 Application to Lidar Temperature Profiles

In the following we apply the wavelet analysis to a temperature profile obtained by the Compact Rayleigh Autonomous Lidar (CORAL). CORAL is a mobile lidar system developed and built by the German Aerospace Center (DLR) and provides temperature measurements from approximately $15\,\mathrm{km}$ to $100\,\mathrm{km}$ altitude. For details see Kaifler and Kaifler (2021). In this work, we analyze a 4 hour long temperature measurement obtained at Río Grande ($53.7°$S, $67.7°$W), Argentina, on the night of 17-18 April 2018. The profile shown in Figure 7a is binned in the vertical to $100\,\mathrm{m}$ resolution with statistical independent values every $900\,\mathrm{m}$. We use this example to demonstrate the CWT analysis and substantiate the points raised in previous sections, as well as show the limitations of the CWT with respect to the wavelet analysis of non-stationary GW signals. The temperature profile in Figure 7a shows significant temperature variability which can be attributed to MWs. Río Grande is in the lee of the Southern Andes and is known to be a hotspot for GWs, in particular MWs (e.g. Hoffmann et al., 2013; Hindley et al., 2020; Rapp et al., 2021; Reichert et al., 2021). The temperature uncertainty peaks at $30\,\mathrm{km}$ and $45\,\mathrm{km}$ where receiving channels are switched, and at altitudes above $90\,\mathrm{km}$. In an initial step, potential GW signals are separated from a thermal larger-scale background. This can be realised in multiple ways. We follow Ehard et al. (2015) and apply a 5th order high-pass Butterworth filter with a



cutoff wavelength of 22 km to the temperature profile in order to separate the temperature background from the GW-induced temperature perturbations. In addition, we compute the theoretically expected upper limit of vertical wavelengths using the Brunt Väisälä frequency determined from the derived temperature background and horizontal winds from ERA5 reanalysis.

After separating the background temperature profile (Fig. 7a) from GW-induced perturbations (Fig. 7b), we apply the polynomial fit as suggested in Section 3.2 in order to derive GW amplitudes. A maximum growth rate of $0.21\,\mathrm{K\,km^{-1}}$ is found at 47 km. The polynomial fit is used to normalize the perturbations before applying the wavelet analysis. Results show short vertical wavelengths and a minimum in GW amplitudes at altitudes of about 25 km. Above, vertical wavelengths become longer and amplitudes increase towards a maximum of 10 K at 75 km. Temperature perturbations are dominated by small scales above 80 km.

The ERA5 profile of absolute winds shows a stratospheric wind minimum around 20-25 km and a maximum at 60 km. The vertical shear of horizontal wind exceeds $3.4\,\mathrm{ms^{-1}km^{-1}}$ between 29 km and 35 km. Assuming that the signal shown in Figure 7b was caused by a MW propagating against the background wind, we can expect an almost linear chirp in the altitude range where the horizontal wind increases linearly with altitude, i.e. between 20 km and 60 km. This is our test case for the application of the CWT.

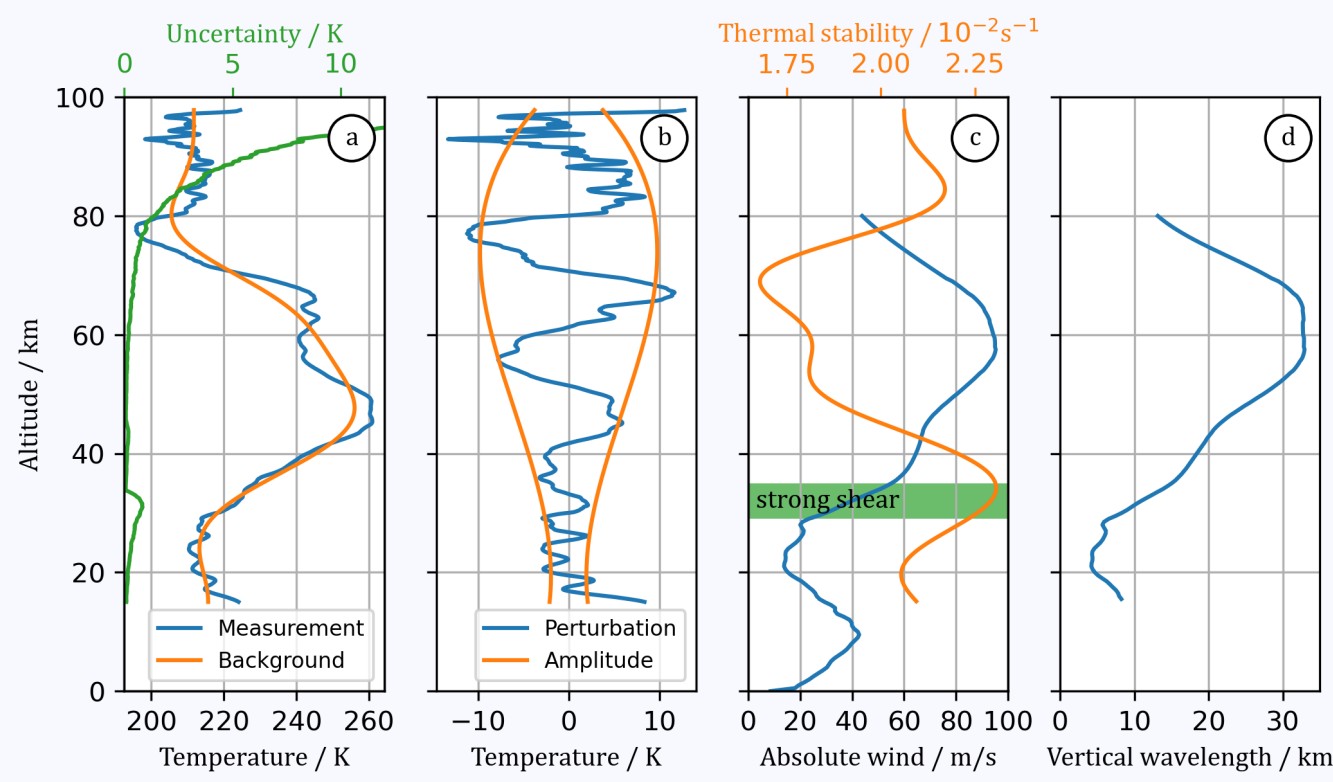

**Figure 7.** (a) Temperature profile obtained by CORAL on the night 17-18 April 2018 (blue), associated temperature uncertainties (green), and determined temperature background (orange). (b) Associated temperature perturbation (blue) and wave amplitude (orange). (c) Absolute wind speed from ERA5 (blue) and stratification (orange). The green region marks altitudes where the wind shear exceeds $3.4\,\mathrm{ms^{-1}km^{-1}}$. (d) Maximum vertical wavelength calculated from ERA5 wind and CORAL background temperature profiles.

### 3.4.1 Analysis of measurement noise

Figure 8 shows the result of our noise analysis based on uncertainties in lidar retrieved temperatures. The procedure is similar to that described in Section 3.3. We generate 5,000 noise profiles with uncorrelated values every $100\,\mathrm{m}$, a mean of $\mu = 0$ and a standard deviation corresponding to the temperature uncertainty at the respective altitude. Subsequently, we compute 5,000 WPS and determine the 99th percentile which is presented in Figure 8. The saw-tooth pattern in the profile of measurement uncertainties (Fig. 7a) is reflected in enhancements of spectral power at corresponding altitudes. If we look at the horizontal stripes with maximum spectral power, we note that these maxima become wider towards longer wavelengths. This is probably due to the fact that the wavelet's localization is weaker at longer wavelengths and noise from distant altitudes contributes to the WPS.





**Figure 8.** 99th percentile computed from 5,000 WPS of lidar measurement uncertainties. The hatched blue region marks the COI.

### 3.4.2 Wavelet analysis of GW-induced temperature perturbations

We now create a WPS in the conventional way in which the amplitudes of the temperature disturbance are not normalized, the order of the wavelet is set to $m_0 = 6$, and no significance levels are determined (Fig. 9a). Furthermore, we create a WPS

based on our best-practice procedure where the amplitudes of the temperature perturbation are normalized, the order of the wavelet is set to $m_0 = 4$, and significance levels are calculated (Fig. 9b). In the conventional WPS we find a maximum at $\lambda_z = 20$ km between 25 km altitude and the top of the profile. However, due to the COI, determined $\lambda_z$ can be considered reliable only between 40 km and 70 km altitude. The maximum below 25 km is found at $\lambda_z = 4$ km. Again, due to the COI, the determined $\lambda_z$ can be considered reliable only between 20 km and 25 km altitude. The conventional WPS shows no other

features at wavelengths $< 20$ km above 30 km altitude. Let us now turn to the best-practice WPS (Fig. 7b). We also find a maximum at $\lambda_z = 20$ km. It is not as extended in altitude as in the conventional case and wavelengths can be determined





reliably between 35 km and 80 km altitude due to the smaller COI. If we follow the maximum down to lower altitudes, we find shrinking wavelengths starting at $\lambda_z = 6$ km at 35 km altitude and reducing to $\lambda_z = 3.5$ km at 20 km altitude. The best-practice WPS shows also no maxima at wavelengths shorter than 20 km at altitudes above 35 km. Only by considering the 99 %

significance level it is possible to diagnose the local maxima at $\lambda_z < 8$ km above 90 km as unreliable.

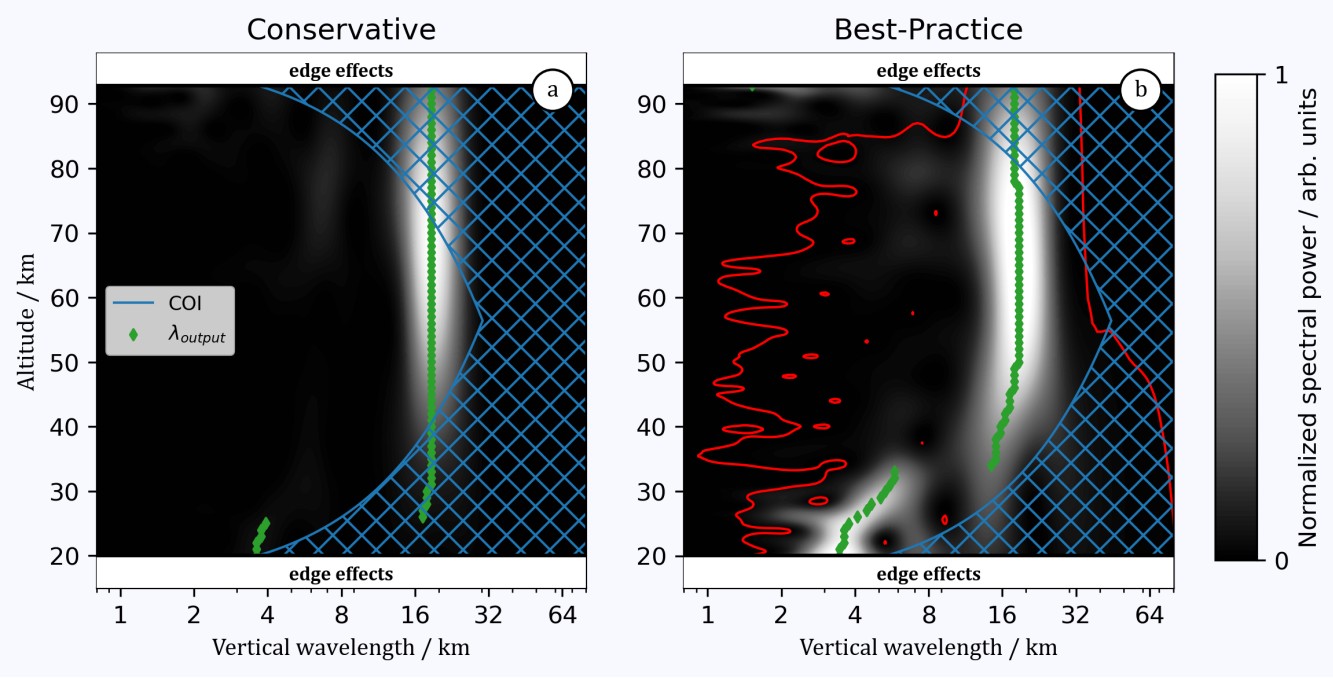

**Figure 9.** (a) WPS of measured temperature perturbations from 17-18 April 2018 for $m_0 = 6$. (b) Same as (a) but following best practices in the computation of the WPS. Green diamonds mark the derived vertical wavelengths, i.e. the maxima in the WPS at each $z$. The red line marks the 99 % significance level. The hatched blue regions mark the COI.

## 4   Discussion

We investigated the choice of wavelet order, amplitude normalization, and determination of significance levels using linear chirps. To a first approximation, i.e. for sufficiently short height regions, it can be assumed that the vertical wavelength of GWs changes linearly. The study of linear chirps using the CWT shows the limitations of the CWT, but also how these limitations

can be extended if necessary and what consequences this has for the interpretation of the results.

In general, multiple GWs can overlap in space and time and one has to investigate not only the global maximum but also local maxima in the WPS to separate individual GWs. This was done, for instance, by Chane-Ming et al. (2000); Rauthe et al. (2008); Baumgarten et al. (2017); Reichert et al. (2021). However, in order to systematically investigate the limitations of wavelet analysis with respect to its ability to separate superimposed GWs, further analyses which are beyond the scope of this

work, are necessary.



### 4.1 The choice of $m_0$

Linear GW theory shows that vertical wavelengths of GWs are proportional to horizontal wind speeds. Therefore, vertical wind shear causes shifts in the vertical wavelength of GWs. Our test cases suggest that the resolvable chirp rate is sensitive to the order of the wavelet (Fig. 3). When the shear region is stronger localized than the wavelet itself, edge effects influence the WPS

and thus the determined wavelengths. We suggest to first inspect background wind and stability in order to make an educated guess of the GW's wavelength shift. Depending on the expected chirp rate, we recommend to use the highest order wavelet possible in the CWT computation since we find that the accuracy of the determined wavelengths decreases for lower order wavelets. As an example, consider a MW signal modulated by a wind shear of $4.5\,\mathrm{ms^{-1}km^{-1}}$. The highest possible wavelet order that should be used to study the signal is $m_0 = 4$, since the accuracy of the determination of the wavelength decreases

significantly when higher orders are used (see Table 3). However, if the wind shear is only $1.5\,\mathrm{ms^{-1}km^{-1}}$, a wavelet order of $m_0 = 8$ should be used, since the accuracy of the determination of the wavelength is better for this order than for lower orders. That in mind, it is very likely that the distribution of vertical wavelengths presented in Reichert et al. (2021) is biased towards shorter wavelengths since they used a fourth order wavelet in their wavelet analysis. On the other hand, Reichert et al. (2021) did not normalize GW signatures, which can have the opposite effect leading to overestimated vertical wavelengths (Fig. 5).

Edge effects arising from weak wavelet localization become a problem in regions where strong wind shear is expected, such as the mid-latitude wintertime lower stratosphere. For the example presented in Figure 7 we find wind shears exceeding $3.4\,\mathrm{ms^{-1}km^{-1}}$ in the lower stratosphere (Fig. 7c) and even values of up to $7\,\mathrm{ms^{-1}km^{-1}}$ are not rare in austral winter in the Southern Andes region. As expected from reanalysis winds, it is the region of strongest shear where the dominant vertical wavelength transitions quickly from $\lambda_z = 7\,\mathrm{km}$ to $\lambda_z = 15\,\mathrm{km}$ in the WPS (Fig. 9). Following the traditional analysis, this

jump might be interpreted as a hint on two distinct and often termed "quasi-monochromatic" wave packets. However, with our new best-practice approach there is evidence that the observed signature reflects a MW undergoing a rapid wavelength shift. As shown in this work, the best choice of $m_0$ depends on the expected wavelength shift and therefore, background wind shear and thermal stability should be investigated before applying wavelet analysis.

### 4.2 GW amplitude normalization

According to linear theory, GW amplitudes increase exponentially with altitude in the absence of dissipation. We demonstrated in Section 2.1 that the exponential variation of GW amplitudes results in spectral leakage of wavelet power to altitudes with smaller GW amplitudes and hence, growing GW amplitudes lead to inaccuracies in determined vertical wavelengths. To our knowledge, no other work has yet investigated this effect even though it appears the effect is known in literature as for example Gisinger et al. (2022) normalized GW signals when comparing lidar measurements and results from a numerical weather

prediction model, and Vadas et al. (2018) scaled temperature perturbations with density. However, they did not investigate systematic differences between WPS of normalized and unnormalized GW signals. In this work, we normalized the GW signals by dividing them by a fourth degree polynomial obtained by a fit to the wave amplitudes. The order of the polynomial should be such that it has as little energy as possible at wavelengths in the spectral range of interest. We found that regardless





of the growth rate it is better to not normalize GW signals as long as the wind shear remains weaker than about $1.5\,\mathrm{ms}^{-1}\mathrm{km}^{-1}$

(Table 4). As the wind shear increases, normalization provides better results. In particular, the wavelength ratios are less scattered (see distributions in Fig. 5). At the same time, however, normalization leads to a systematic underestimation of the vertical wavelength, as already shown in the case of constant amplitudes (Table 3). Again, we suggest to inspect background wind profiles before applying wavelet analysis and normalize GW signals when wind shears larger than $1.5\,\mathrm{ms}^{-1}\mathrm{km}^{-1}$ are expected.

Since the wind shear in the case study (Fig. 7) easily exceeds the $1.5\,\mathrm{ms}^{-1}\mathrm{km}^{-1}$, we normalized the temperature perturbations before applying the CWT. It is this additional step which allowed us to capture the evolution of the MW (see Fig. 9b), revealing an approximate doubling in vertical wavelength from $\lambda_z = 7\,\mathrm{km}$ to $\lambda_z = 15\,\mathrm{km}$ at approximately 35 km altitude.

### 4.3 Significance levels in wavelet power spectra

Chane-Ming et al. (2000); Werner et al. (2007); Reichert et al. (2021) used temperature amplitudes to determine whether signals

of interest are reliable. By doing so they made the implicit assumption of a flat noise spectrum. This may be approximately true for certain spectral regions, but generally this assumption cannot be made for real measurement data. For example, using wavelet analysis to investigate the noise in lidar data, we were able to show that the spectral amplitudes increase toward long vertical wavelengths, revealing the characteristics of red noise (Fig. 8). Therefore, even if the noise level is low at a specific altitude, a large scale signal could be potentially not significant at this very same altitude due to higher noise levels at distant

altitudes. We argue that it is crucial to compute significance levels as described in Section 3.3 in order to reliably determine wavelengths. In our case study all maxima in the WPS are significant (Fig. 9).

### 5 Summary and Conclusion

We studied the determination of vertical wavelengths based on wavelet analysis using first artificial test signals and later lidar temperature measurements. We discussed the treatment of measurement uncertainties, the impact of GW amplitudes increasing

with altitude, and the influence of chirps that arise due to the vertical shear of horizontal wind. Following tests with artificially created data, we presented a recipe which aims to minimize the influence of edge effects in wavelet analysis. For the analysis of lidar data, we suggest to first inspect atmospheric background variables such as horizontal wind speed and thermal stability and then, depending on the particular atmospheric conditions, choose the wavelet order which is most suitable for analyzing the data. For measurements taken at a mid-latitude site like Río Grande ($54°\,\mathrm{S}$) in winter we set $m_0 = 4$. This choice has the advan-

tage that for one the admissibility condition is still met and second, chirps due to wind shears of up to $\partial_z u = 4.5\,\mathrm{ms}^{-1}\mathrm{km}^{-1}$ can be resolved. In addition, the $e$-folding length is smaller, resulting in weaker edge effects. Second, prior to the wavelet analysis, GW amplitudes usually should be normalized in order to prevent spectral leakage. In this work, a 4th degree polynomial fit was found to be a suitable normalization method. Third, the noise characteristic of the instrument is used to compute a noise-WPS which in turn is used to determine significance levels.






In the following, we give step-by-step instructions on how to analyze lidar data for GWs.

1. **Separation of background and perturbation**

   Apply a 5th order Butterworth filter in the vertical. In a first substep, the cutoff is set to the maximum wavelength that can be expected from theoretical considerations, i.e. for MWs $\lambda_z = 2\pi\frac{u}{N}$. This cutoff is usually too large at first and is set in a second substep to the maximum wavelength that results from the *wavelength determination*. The Butterworth filter and other approaches are extensively discussed in Ehard et al. (2015).

2. **Amplitude Normalization**

   Compute the running RMS of the temperature perturbation over a window size that is equal to the cutoff wavelength of the Butterworth filter. Fit a polynomial to the running RMS and use the result to normalize the perturbations. The polynomial fit should capture only large scales and the degree of the polynomial should be chosen such that the spectral power in the wavelength range containing GW signals remains approximately unaffected.

3. **Wavelength determination**

   Compute the WPS of the normalized temperature perturbation profile. Create a profile of vertical wavelength by identifying maxima in the WPS at each altitude.

4. **Noise WPS computation**

   Generate 5,000 Gaussian noise profiles with a standard deviation given by the measurement uncertainty as function of altitude. Compute the WPS of these profiles and determine the percentile of the WPS that is associated with the desired significance level.

5. **Assessment of significance levels**

   Consider only vertical wavelengths in regions outside the COI and where the desired significance level is reached. Disregard the lower- and uppermost $\sim 5\,\mathrm{km}$ in altitude because of edge effects from the Butterworth filtering as well as the *amplitude normalization*.

The presented limitations of wavelet analysis and work-arounds can be easily applied to temperature or wind profiles obtained, for instance, by other lidars and also by radars, radiosondes, and satellites. In essence, when choosing the wavelet transform for the investigation of GW signals in vertical profiles, one first must come up with an educated guess on expected wavelength shifts and amplitude growth. We found that the evolution of these two quantities determines in large parts the feasibility of the wavelet analysis.

*Code and data availability.* Lidar data as netCDF file and IDL code for the wavelet analysis is made publicly available at Zenodo.org.



*Author contributions.* RR developed the method, carried out all data analysis, and wrote the manuscript. NK and BK provided the CORAL
370    data and revised the manuscript.

*Competing interests.* The authors declare that they have no conflict of interest.

*Acknowledgements.*



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
