# Peer review of "Limitations in Wavelet Analysis of Non-Stationary Atmospheric Gravity Wave Signatures in Temperature Profiles"

_Atmospheric Measurement Techniques, 2023_

## Author Comment (AC2)

a) Temperature perturbations observed by CORAL on the night of 21-22 May 2018. Contour lines indicate temperature perturbations from ERA5. b) Obtained CORAL (red) and ERA5 (black) temperature profiles at 00UTC. c) Zonal crosssection of ERA5 temperature perturbations. d) Meridional crosssection of ERA5 temperature perturbations. Dashed lines in c) and d) mark the CORAL location.

---

## Author Response (AR1)

Dear Editor,

We are grateful for all the feedback we received from you and the two referees. We hope that we accounted for all comments in the second version of our manuscript and it can be published soon. We have revised our manuscript carefully. Since it was suggested by referee #1 we added a figure (now Figure 1) to the manuscript which illustrates three Morlet wavelets with the same scale but different orders. In addition, we show their representation in the distance-wavelength-space and Fourier space in order to highlight the localization in each space as function of the order.

We have revised the text at the following lines:

L100-101: "Three Morlet wavelets with orders of 4, 6, and 8 are illustrated in Figure 1."

L126: "A scale of s=5km is equivalent to a wavelength of 3.9km for an order of 4 and 7.6km for an order of 8 (see Fig. 1)."

L223: "[…] we analyze a temperature profile obtained at Río Grande (53.7°S, 67.7°W), Argentina on the night of 21 May 2018 00UTC."

L232: We change the cutoff from 22km to 20km.

L235: "[…] and juxtapose the measured vertical wavelength using our best practice (Fig. 8d)."

L235-236: "Reanalysis data is spectrally truncated at wavenumber T21 in order to define a synoptic-scale background (e.g. Reichert et al., 2021)."

L238: We change the maximum growth rate from 0.21K/km at 47km to 0.82K/km at 36km.

L241: We change the maximum from 10K at 75km to 20K at 55km.

We erased the following:
L242-245: "The ERA5 profile of absolute winds shows a stratospheric wind minimum around 20-25km and a maximum at 60km. The vertical shear of horizontal wind exceeds 3.4m/s/km between 29km and 35km. Assuming that the signal shown in Figure 8b was caused by a MW propagating against the background wind, we can expect an almost linear chirp in the altitude range where the horizontal wind increases linearly with altitude, i.e. between 20km and 60km."

And replaced it with:
L242-245: "The ERA5 profiles of zonal and meridional wind show a rather steady increase of wind speeds between 20km and 50km. The vertical shear of horizontal wind exceeds 3.4m/s/km between 32km and 37km. At this altitude, we find a discontinuity in the profile of measured vertical wavelengths. Computed and measured vertical wavelengths agree quite well below 35km but differ by up to a factor of two above 35km."

We have rewritten subsection 3.4.2
"We now create a WPS in the conventional way in which the amplitudes of the temperature disturbance are not normalized, the order of the wavelet is set to $m_0=6$, and no significance levels are determined (Fig. 10a). Furthermore, we create a WPS based on our best-practice procedure where the amplitudes of the temperature perturbation are normalized, the order of the wavelet is set to $m_0=4$, and significance levels are calculated (Fig. 10b).

In the conventional WPS we find only little variation in the vertical wavelength with values ranging from 12.7km to 16.4km. Values decrease from the upper and lower edge of the profiles towards 65km altitude. Due to the COI, determined Lz can be considered reliable only between 35km and 75km altitude. The conventional WPS shows no other interesting features.
Let us now turn to the best-practice WPS (Fig. 10b).
Similar to the conservative case, we find an extended altitude region from 30km to 80km with only little variation in the vertical wavelength with values of 11.1km to 15.0km which are smaller than the values found in the conservative case. This agrees with the results from our sensitivity study (Section 3.1). In contrast to the conventional case we are able to identify maxima in the WPS now at vertical wavelengths from 4.7km at 20km altitude to 9.8km at 30km altitude and vertical wavelengths in the order of 6km to 7km above 80km altitude."

L297: We changed "Lz=7km" to "Lz=10km"

We added:
L301-304: "ERA5 temperature perturbation fields and co-located OH-airglow imagery provide more evidence that the MW observed by CORAL propagates steeply within the lidar's field of view. On the other hand, the difference between computed and measured vertical wavelength (Fig. 8d) could be an indication for an obliquely propagating MW. After all, this work is of methodological nature and the geophysical interpretation of the results is not in our focus."

L325: "[…] revealing an increase of vertical wavelength from Lz=10km to Lz=15km at approximately 32km altitude."

L352: "[…] mid-frequency MW"

To your 1):
We addressed the angle between wind direction and wave propagation direction by pointing out that we see a difference in the measured and computed vertical wavelength which could be a hint on lateral propagation.

To your 2):
The focus of this work is to determine the limitations of the continuous wavelet transform. One of the aspects was to investigate which chirp rates, i.e. frequency shifts can be resolved using the CWT. In order to learn more about the implications of this study, we relate the linear chirps we define in Section 2.2 to horizontal wind shear using a) the mid-frequency approximation, b) assuming we are dealing with mountain waves, and c) assuming that wind and wave propagation direction are constant.

To your 3):
Indeed, additional ERA5 analysis has shown that the nightly mean profile we used previously from the 17-18 April 2018 showed very likely a superposition of two mountain waves that propagated obliquely through CORAL's field of view.
Therefore, we decided to show another temperature profile from 21 May 2018 (see Figure 8a, b in the manuscript). ERA5 analysis shows that mountain waves are excited close by the lidar location and propagate steeply (see Figure 1 c, d).

[Figure]

*Figure 1 Temperature perturbations observed by lidar (a), temperature profile at 00UTC (b), zonal cross section of ERA5 temperature perturbations (c), and meridional cross section of ERA5 temperature perturbations (d). Dashed lines indicate the lidar location.*

To your 4):
We refrain from performing idealized numerical simulations in order to improve the interpretation of the results from section 3.4. As stated before, the geophysical interpretation is not the focus of this work.

**Author's response to reviewer #1**

This paper investigates the sensitivity of the wavelet transform of atmospheric vertical profiles to the altitudinal variations of the wavelengths and amplitudes of internal gravity waves. The choice of the order of the Morlet mother wavelet is also discussed. The authors recommend normalizing gravity wave signals before applying wavelet analysis. The analyses presented are helpful for researchers using wavelet transform for atmospheric studies. The paper will become acceptable for publication after answering the following issues.

- Given that the sensitivity of the WPS to the order of the Morlet wavelet is investigated, plots of the wavelet for m0 = 4, 6, 8 should be given.

**Even though it seems a bit redundant to show another illustration of a Morlet wavelet, the suggestion is certainly easy to implement and will not change the manuscript for the worse.**

- Eq.(7): It should be mentioned based on this relation that the COE is wider for larger m0 so that the readers can understand the cause of the different COE for different m0 in the subsequent WPSs.

**We thank the referee for that comment and will include the following statement: Larger orders result in a larger extend of the Morlet wavelet and hence to a more extended COI in the WPS.**

- Different units are connected without spaces, like "ms" and "Kkm", in many places. Insert spaces like "m s" and "K km."

**We thank the referee for that comment and will correct for that.**

- Figure 3: Why are there many experiments for each combination of the wind shear and wavelet order? There is a description "no additive noise", so there seems to be no Monte Carlo processes. Which parameter is different between those experiments?

**There is no Monte Carlo process involved at this place. Line154-155: "Figure 3 illustrates the distributions of wavelength ratios derived from the lowermost 50 km of the simulated altitude range […]" With a vertical discretization of dz=0.1km, we obtain 500 wavelength ratios that are plotted as histograms for each wind shear and wavelet order.**

- l.177: "Equ,4" might be better as "Eq. 4" or "Equation 4. "

**Will be changed as suggested.**

- l.194: How the "growth rates" are applied is unclear. I guess the formulation "a(z) = 1.0 K + z K km-1" given in l.139 is modified as "a(z) = 1.0 K + 0.1*z K km-1", "a(z) = 1.0 K + 1*z K km-1", and "a(z) = 1.0 K + 10*z K km-1". More explanations are needed.

**That is absolutely correct. We apologize for that oversight and will correct the according expressions in line 139 and table 2.**

- Figure 5: Similar to Fig.3, I do not understand why many experiments exist for each combination of the normalization, wind shear, and growth rate.

**We apologize for the misunderstanding. In fact, we have not mentioned that the distributions are only considering the lowermost 50km as it is the case in figure 3.**

- Figure 7: Panel d is not mentioned in the text. Moreover, how this vertical wavelength was obtained is not explained.

**We thank the referee for this comment. We will change the following lines as following:**

**Line 230-231: "In addition, we compute the theoretically expected upper limit of vertical wavelengths according to $\lambda_z = 2\pi \frac{u}{N}$ using the Brunt Väisälä frequency determined from the derived temperature background and horizontal winds from ERA5 reanalysis (Fig. 7d).**

**Line 238: "The ERA5 profile of absolute winds (Fig. 7c) shows […]"**

**Line 240-241: "[…] we can expect an almost linear chirp in the altitude range where the horizontal wind increases linearly with altitude, i.e. between 20 km and 60 km (see Fig. 7d)."**

- l.260: "(Fig. 7b)": Is this a typo for "(Fig. 9b)"?

**Thank you. That is a typo indeed. It should be (Fig. 9b).**

- l.262: "due to the smaller COE": Is this smaller COE due to the smaller m0? If so, mention it.

**We will add the following to line 262: […] which is due to the usage of a smaller order ($m_0 = 4$).**

- l.272: An additional example of wavelet analysis for conditions where multiple waves overlap:

Mori, R., Imamura, T., Ando, H., Häusler, B., Pätzold, M., & Tellmann, S. (2021). Gravity wave packets in the Venusian atmosphere observed by radio occultation experiments: Comparison with saturation theory. Journal of Geophysical Research, 126, e2021JE006912. https://doi.org/10.1029/2021JE006912

**We will include the reference.**

- l.277: For the vertical wavelength to be proportional to the horizontal wind speed, the wave must be a topographically generated gravity wave (mountain wave). This should be mentioned. If the paper focuses on mountain waves, the title of the paper is not appropriate since mountain waves are stationary.

**We have mentioned mountain waves at various places in the manuscript:**

**Line 7: "[…] we apply CWT to test mountain wave signals […]"**

**Line 40: "The vertical wavelength of stationary mountain waves (MW) […] is approximately given as $\lambda_z = 2\pi \frac{u}{N}$ […]"**

**Line 224: "The temperature profile in Figure 7a shows significant temperature variability which can be attributed to MWs."**

**Line 239-241: "Assuming that the signal […] was caused by a MW propagating against the background wind, we can expect an almost linear chirp in the altitude range where the horizontal wind increases linearly with altitude […]."**

**Line 283: "[…] consider a MW signal […]"**

**We want to clarify the title. The manuscript is about non-stationary signatures of atmospheric gravity waves and not about signatures of non-stationary atmospheric gravity waves. The term non-stationary refers to a change of statistical properties such as variance and autocorrelation of the data under consideration over time/length. Even though linear mountain waves are a stationary physical feature in the atmosphere, their signal in the vertical is clearly non-stationary in the statistical sense.**

**Author's response to reviewer #2**

The manuscript "Limitations in Wavelet Analysis of Non-Stationary Atmospheric Gravity Wave Signatures in Temperature Profiles" by Reichert et al. is a nice discussion of the limitations of the Continuous Wavelet Transform (CWT) that is often used to analyze non-stationary signals. First, idealized linear chirp signals are used as test signals for optimizing the method. Then the method is applied to a temperature profile observed by the CORAL lidar in South America. Recommendations are made regarding the order of the Morlet wavelet. Further, the authors suggest to normalize observed temperature profiles before performing a wavelet analysis.

Overall, the manuscript is well written and fits into the scope of AMT.

Therefore the manuscript is recommended for publication after addressing my minor but important comments as detailed below.

My main comment is that the authors should point out more clearly that the cases discussed in the paper are focused on a very specific meteorological situation (midlatitude winter conditions where wind reversals do not occur at altitudes below 80km), while conditions can be much more complicated in the general case.

SPECIFIC COMMENTS:

(1) l.53-58:

As in your paper you are investigating cases of wind shear, you should mention more explicitly that in the presence of wind shear parts of the gravity wave spectrum will dissipate which can lead to rapid decrease of gravity wave amplitudes with altitude. Therefore, exponential growth often occurs, but does not hold in every situation.

**That is correct and we thank the reviewer for that comment. We have altered l.53-57 in the following way:**

**"Secondly, according to linear GW theory, not only can vertical wavelengths change rapidly, but the amplitudes of GWs also vary with altitude. Generally, amplitudes increase exponentially with altitude enforced by conservation of energy and decreasing air density. However, when thermal or dynamical instability is reached, wave dissipation occurs, causing GW amplitudes to decrease above the breaking altitude. This variation in GW amplitude may lead to an undesirable shift in the localization of the wavelet during the computation of the CWT."**

You should also mention that another method to compensate for the growth of gravity wave amplitudes would be the investigation of gravity wave potential energy per unit mass. In this way, not only density variations but also variations of the buoyancy frequency N and of the temperature with altitude are accounted for (Strelnikova et al., 2021).

Strelnikova, I., Almowafy, M., Baumgarten, G., Baumgarten, K., Ern, M., Gerding, M., and Lubken, F.-J.:

Seasonal cycle of gravity wave potential energy densities from lidar and satellite observations at 54N and 69N,

J. Atmos. Sci., 78, 1359-1386, https://doi.org/10.1175/JAS-D-20-0247.1, 2021.

**We disagree with this statement. It is correct, that the GW potential energy per unit mass is a function not only of density but also of ambient temperature and thermal stability. However, the mere investigation of GW potential energies is not an adequate method to compensate for the growth of GW amplitudes.**

(2) l.65-68:

It is important to mention here that close to a critical wind level gravity wave amplitude behavior can be very complicated, and the assumption of exponentially growing amplitudes due to density

decrease may no longer hold: On the one hand, close to a critical level, as long as the gravity wave is not saturated, the gravity wave amplitude will grow stronger than suggested by the density decrease in order to compensate for the reduction in vertical wavelength caused by the Doppler shift. On the other hand, even closer to the critical level, the gravity wave amplitude will decrease because the wave saturates which limits the wave amplitude.

For a discussion see, for example, Ern et al., 2014, their Section 3.1

Ern, M., F. Ploeger, P. Preusse, J. C. Gille, L. J. Gray, S. Kalisch, M. G. Mlynczak, J. M. Russell III, and M. Riese (2014), Interaction of gravity waves with the QBO: A satellite perspective, J. Geophys. Res. Atmos., 119, 2329-2355, doi:10.1002/2013JD020731.

This more complex situation in the general case would be a justification for the method you propose later in lines 186-192 which allows for both increasing and decreasing amplitudes.

**We wrote the paragraph in l.65-72 to highlight the importance to determine vertical wavelengths of GWs and not their amplitudes. The approximation of amplitudes is just a necessity in order to determine proper vertical wavelengths. We agree, that the behavior of GW amplitudes close to a critical wind level can be very complicated, but since we use a 4$^{th}$ order polynomial to approximate GW amplitude variation with altitude, the proposed method in l.190-195 results obviously only in a rough estimation of GW amplitudes. To put it in a nutshell, since we are not interested in the precise determination of GW amplitudes, we will not lead the reader at this point in the manuscript down the wrong path.**

(3) l.277: This is not correct. "Proportionality" is given only for orographic gravity waves, and even in this case proportionality applies only to the wind component parallel to the gravity wave horizontal wave vector. In case of midfrequency gravity waves vertical wavelength is proportional to intrinsic phase speed, and there is a "linear" relationship between vertical wavelength and the wind speed parallel to the gravity wave horizontal wave vector.

**We thank the reviewer for pointing this out. We have altered l.280 in the following way:**

**"Linear GW theory shows that vertical wavelengths of GWs are a function of horizontal wind speed."**

TECHNICAL COMMENTS:

l.50: Please state that m_0 in Table 1 is the order of the Morlet wavelet.

This is defined later in the paper, but needs to be introduced already here.

**We add "order" in front of m_0 in the headline of table 1.**

l.141: (Fig.1cd). -> (Fig.1c, d).

**We change that as suggested.**

Caption of Fig.1 and elsewhere: same problem, please separate labels of the different panels by commas

**We change that as suggested.**

l.385, reference Chane-Ming et al.: Journal title is sufficient, publisher (Springer) can be omitted

**We change that as suggested.**

l.393: Please check title of reference Ehard et al.

**We thank the reviewer for pointing that out. We change that accordingly.**

l.405, reference Ge: Journal title is sufficient, publisher (Copernicus) can be omitted

**We change that as suggested.**

l.385, reference Chane-Ming et al.: Journal title is sufficient, publisher (Springer) can be omitted

---

## Author Response (AR2)

Dear editor, dear anonymous referees,

We are grateful for the comments we received and answer to them in the following. We have copied your comments below and our answers are printed in blue.

Anonymous Referee #1

The reviewer's concerns about the original manuscript have mostly been solved, and the revised manuscript is considered acceptable for publication. I have the following minor comments, but I want to leave these points to the authors.

- Now I understand that the many data points for each wind shear and wavelet order combination in Figure 4 correspond to different altitudes from 0 to 50 km with an interval of 0.1 km. This point should be clearly described in the text because it is unclear in the current description.

We have added the number of wavelength ratios to make this point a bit more clear: "Figure 4 illustrates the distributions of **500** wavelength ratios derived from the lowermost 50 km of the simulated altitude range and Table 3 lists the corresponding median deviations as well as interquartile ranges (IQR)."

- I came to understand the meaning of 'Non-Stationary' in the title, but I still recommend changing it to avoid a potential misunderstanding. Rephrasing it to 'Non-uniform' will be more appropriate.

We understand the distinction between "non-stationary" and "non-uniform" as the former refers to variable statistical properties in time while the latter refers to variable statistical properties in space. However, we refrain from rephrasing the title and exchanging "non-stationary" by "non-uniform" multiple times in the manuscript. From our perspective the dimension, i.e. time and space, is not of importance for our analysis. The suggestions we make are applicable to timeseries as well as vertical profiles.

Anonymous Referee #2

The revised manuscript "Limitations in Wavelet Analysis of Non-Stationary Atmospheric Gravity Wave Signatures in Temperature Profiles" by Reichert et al. reads much better now. Still, there are a few minor corrections required before publication in AMT.

SPECIFIC COMMENTS:

l.174: In Eq.(10) you are using $m\_o$ as the vertical wavenumber of the GW.
However, in this equation the vertical wavenumber $m$ should be an independent parameter.
Later on, you should state that in Eq.(11) you choose $m\_o=m$ to illustrate the effect of biases that are introduced if no amplitude scaling is applied.

We thank the reviewer for this comment and have adapted Eq.(10). In addition, we added the half sentence after Eq.(12): "[…] when choosing $m=\frac{m_0}{s}$, i.e. the Morlet wavelet's vertical wavenumber is equal to the GW's vertical wavenumber."

l.175: initial temperature -> initial GW temperature amplitude

We thank the reviewer for this comment and have added "amplitude".

l.303: The statement "obliquely propagating MW" is too vague!
You should explicitly mention that above 35km possibly not the same GW is observed as below 35km because the vertical wavelength no longer matches the expectations for a mountain wave.

We have altered lines … in the following way:
"Following the traditional analysis, this jump might be interpreted as a hint on two distinct and often termed "quasi-monochromatic" wave packets. The difference between computed and measured vertical wavelength (Fig. 8d) could be an indication for two different wave packets, one below and one above ~32km. However, with our new best-practice approach there is evidence that the observed signature reflects a MW undergoing a rapid wavelength shift. ERA5 temperature perturbation fields and co-located OH-airglow imagery (both not shown) provide more evidence that the MW observed by CORAL propagates steeply within the lidar's field of view. After all, this work is of methodological nature and the geophysical interpretation of the results is not in our focus."

TECHNICAL COMMENTS:

l.100: where -> where, in our case,

We changed that as suggested.

l.121: extend -> extent

We changed that as suggested.